# High-lying valley-polarized trions in 2D semiconductors

Kai-Qiang Lin [1]✉, Jonas D. Ziegler[2], Marina A. Semina[3], Javid V. Mamedov[4], Kenji Watanabe[5], Takashi Taniguchi[6], Sebastian Bange[1], Alexey Chernikov[2], Mikhail M. Glazov[3,4] ✉ & John M. Lupton[1]

Optoelectronic functionalities of monolayer transition-metal dichalcogenide (TMDC) semiconductors are characterized by the emergence of externally tunable, correlated many-body complexes arising from strong Coulomb interactions. However, the vast majority of such states susceptible to manipulation has been limited to the region in energy around the fundamental bandgap. We report the observation of tightly bound, valley-polarized, UV-emissive trions in monolayer TMDC transistors: quasiparticles composed of an electron from a high-lying conduction band with negative effective mass, a hole from the first valence band, and an additional charge from a band-edge state. These high-lying trions have markedly different optical selection rules compared to band-edge trions and show helicity opposite to that of the excitation. An electrical gate controls both the oscillator strength and the detuning of the excitonic transitions, and therefore the Rabi frequency of the strongly driven three-level system, enabling excitonic quantum interference to be switched on and off in a deterministic fashion.

Transition-metal dichalcogenide (TMDC) monolayers are known to host tightly bound band-edge excitons[1–3], as well as a variety of more elaborate many-body species such as exciton complexes in the presence of the Fermi sea of free carriers (known as trions and Fermi polarons)[4–6] and excitonic molecules[7–15]. More recently, bound high-lying excitons (HXs), involving electrons from the upper conduction band (in particular, CB+2) and holes from the top-most valence band, have been observed[16]. Even though these high-lying excitons appear at almost twice the energy of the band-edge exciton (X), they exhibit a particularly narrow linewidth that is comparable to that of band-edge excitons. Intriguingly, *GW*-BSE calculations show that the HX consists of an electron originating predominantly from a downwards-curved conduction band, i.e. a negative effective mass electron[16]. Although trion formation is well-known for the band-edge excitons in two-dimensional semiconductors, it has so far remained unclear whether such states can also form from these more exotic excitonic species.

Coincidentally, in monolayer $WSe_2$, these high-lying excitons appear at around twice the band-edge exciton transition energy, giving rise to a degenerate atom-like three-level system that allows for a pronounced quantum interference phenomenon to occur in optical second-harmonic generation (SHG)[16–19]. Combining such quantum interference, which generally occurs in discrete multilevel systems such as atomic vapors, crystal defects, or ions, with electronic device functionality has been a long-standing goal. An exciton-based three-level system promises the advantage of facile integration into electronic devices and potentially offers unique control over quantum interference through electrical gate signals. However, such control remains a major conceptual challenge and has not been developed yet.

[1]Department of Physics, University of Regensburg, 93053 Regensburg, Germany. [2]Dresden Integrated Center for Applied Physics and Photonic Materials and Würzburg-Dresden Cluster of Excellence ct.qmat, Technische Universität Dresden, 01062 Dresden, Germany. [3]Ioffe Institute, 194021 St. Petersburg, Russia. [4]National Research University, Higher School of Economics, 190121 St. Petersburg, Russia. [5]Center for Functional Materials, National Institute for Materials Science, Tsukuba, Ibaraki 305-004, Japan. [6]International Center for Materials Nanoarchitectonics, National Institute for Materials Science, Tsukuba, Ibaraki 305-004, Japan. ✉e-mail: kaiqiang.lin@ur.de; glazov@coherent.ioffe.ru

Here, we demonstrate experimentally and theoretically that such high-lying trions with negative-mass electrons can indeed form. We probe these charged HXs in a monolayer $WSe_2$ transistor, where the charge carrier density can be continuously controlled via the gate voltage. We generalize these observations by demonstrating that similar features exist in a monolayer $MoSe_2$ transistor. In contrast to the neutral HX, high-lying trions show pronounced helicity following a valley polarization selection rule that is distinct from those of the band-edge transitions. In addition to the bright HX trion, a dark charged *p*-like HX is identified by the signature of excitonic quantum interference in SHG. Finally, we demonstrate robust control of excitonic quantum interference by the gate voltage.

## Results

### Electrical tuning of high-lying trions in monolayer $WSe_2$

Figure 1a illustrates two conceivable configurations of high-lying trions in the vicinity of the K-points in momentum space: the photoexcited electron in the high-energy CB+2 band, and the photoexcited hole in the top valence band (VB) forming a negative (positive) trion with an additional resident electron (hole) at the band edge. To probe these trions, a gate-tunable device is needed to control the charge carrier density, i.e., the doping, in the monolayer. Figure 1b illustrates the monolayer $WSe_2$ transistor device, a microscope photograph of which is shown in Fig. 1c. The gate-voltage dependence of photoluminescence (PL) from band-edge excitons is first measured to characterize the control of doping in monolayer $WSe_2$. Figure 1d reproduces the characteristic features of the well-documented band-edge excitonic species, such as the negatively charged A exciton singlet ($X_S^-$) and triplet ($X_T^-$), the positively charged A exciton ($X^+$), the biexciton ($XX_D$), the intervalley dark exciton ($X_D$), and the charged dark excitons ($X_D^+$ and $X_D^-$)[20–22]. With a thin graphite flake (few-layer graphene) as the top gate, such gate-voltage dependent measurements are completely reversible without noticeable hysteresis, as

discussed in Supplementary Note 1 and Supplementary Fig. 1. From the appearance of the neutral exciton species such as the neutral dark exciton and biexciton in Fig. 1d, we can identify the charge neutrality point centered around −0.15 V.

Resonant pumping of the A exciton followed by Auger-like exciton-exciton annihilation selectively promotes electrons to the high-lying conduction band near the ±K-points in momentum space, forming a high-lying exciton HX[16,23,24]. Supplementary Fig. 2 shows a pump power dependence of UPL supporting the Auger-like process. Figure 1e shows the gate-voltage dependence of upconverted PL (UPL) in the energy range of 3.2 to 3.5 eV under continuous-wave (CW) excitation at 1.724 eV, i.e., at about twice the excitation energy. Figure 1f shows example UPL spectra at gate voltages of +0.2 V (negatively charged), −0.15 V (neutral), and −0.5 V (positively charged). In the neutral regime, the UPL spectrum of the HX (black line) shows a characteristic phonon progression, as reported previously[16], along with a sharp spectral feature at 3.448 eV, which arises due to the CW SHG of the incident laser[25]. The zero-phonon line of the HX is marked as $HX_0$. Strikingly, new peaks emerge under both positive and negative doping (Fig. 1e). In the electron-doping regime, a new peak appears 43 meV below the $HX_0$. In the hole-doping regime, a peak emerges 35 meV below the $HX_0$. These two peaks have narrow linewidths resembling those of the HX and are therefore assigned to the negatively ($HX^-$) and positively ($HX^+$) charged HX. As shown in Fig. 1e and Supplementary Fig. 1, gate voltages of +0.2 V and −0.5 V correspond to a low doping regime where the Fermi level is close to the band edges to within 1 meV[26]. The offsets of the trion transitions observed with respect to $HX_0$ are therefore attributed to their binding energies. As listed in Table S1, these binding energies are, on average, 1.4 times as large as those of band-edge (A exciton) trions, in line with the fact that the HX binding energies, calculated from *ab initio GW*-BSE theory, are about 1.3 times as large as the A-exciton binding energy[16]. The negatively charged HX has larger binding energy than the positively

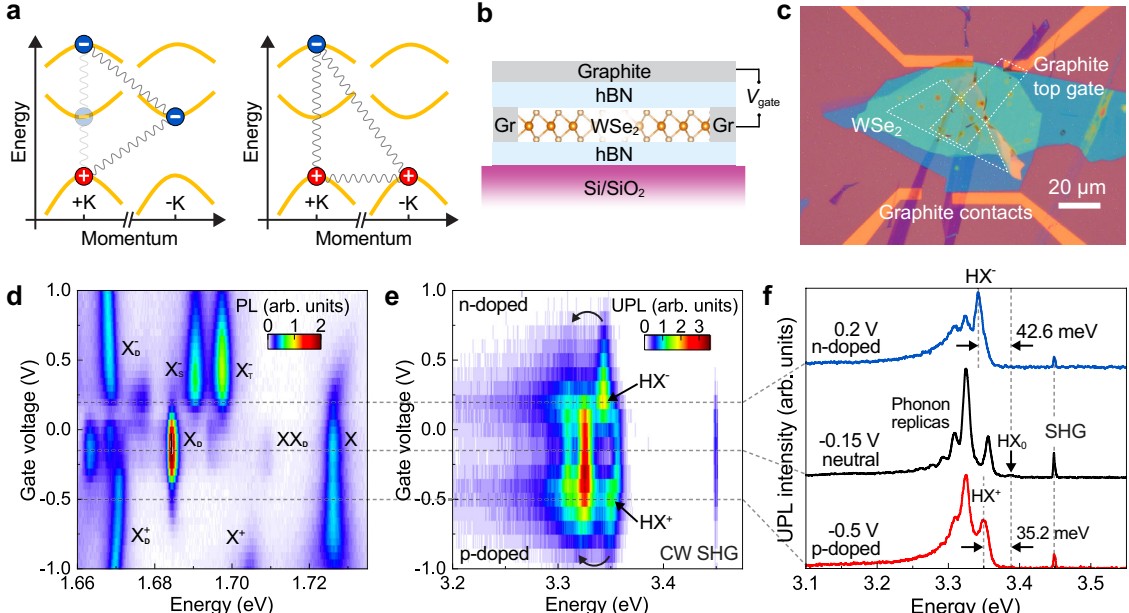

**Fig. 1 | High-lying trions in monolayer $WSe_2$. a** Illustration of negatively and positively charged high-lying excitons (HX), composed of an electron from a high-lying conduction band with negative effective mass, a hole from the first valence band, and an additional charge from a band-edge state. The semi-transparent electron and curved lines on the left illustrate an alternative structure of the negatively charged HX, where the electrons are in different valleys. **b, c** Schematics and microscope photograph of the transistor device. The $WSe_2$ monolayer is encapsulated between thin hBN layers and connected to pre-pattern gold

electrodes via graphite flakes. The uppermost graphite flake covering monolayer $WSe_2$ serves as the top gate. **d** Photoluminescence (PL) of band-edge excitons (X) at 488 nm excitation, measured as a function of gate voltage. **e** Upconverted PL (UPL) of the HX at 719.2 nm (1.724 eV) excitation, measured as a function of gate voltage. The features attributed to the negatively charged HX and to the positively charged HX are marked as $HX^-$ and $HX^+$, respectively. **f** Sample UPL spectra at gate voltages of 0.2 V (electron doping), −0.15 V (neutral), and −0.5 V (hole doping). All measurements were performed at 5 K.

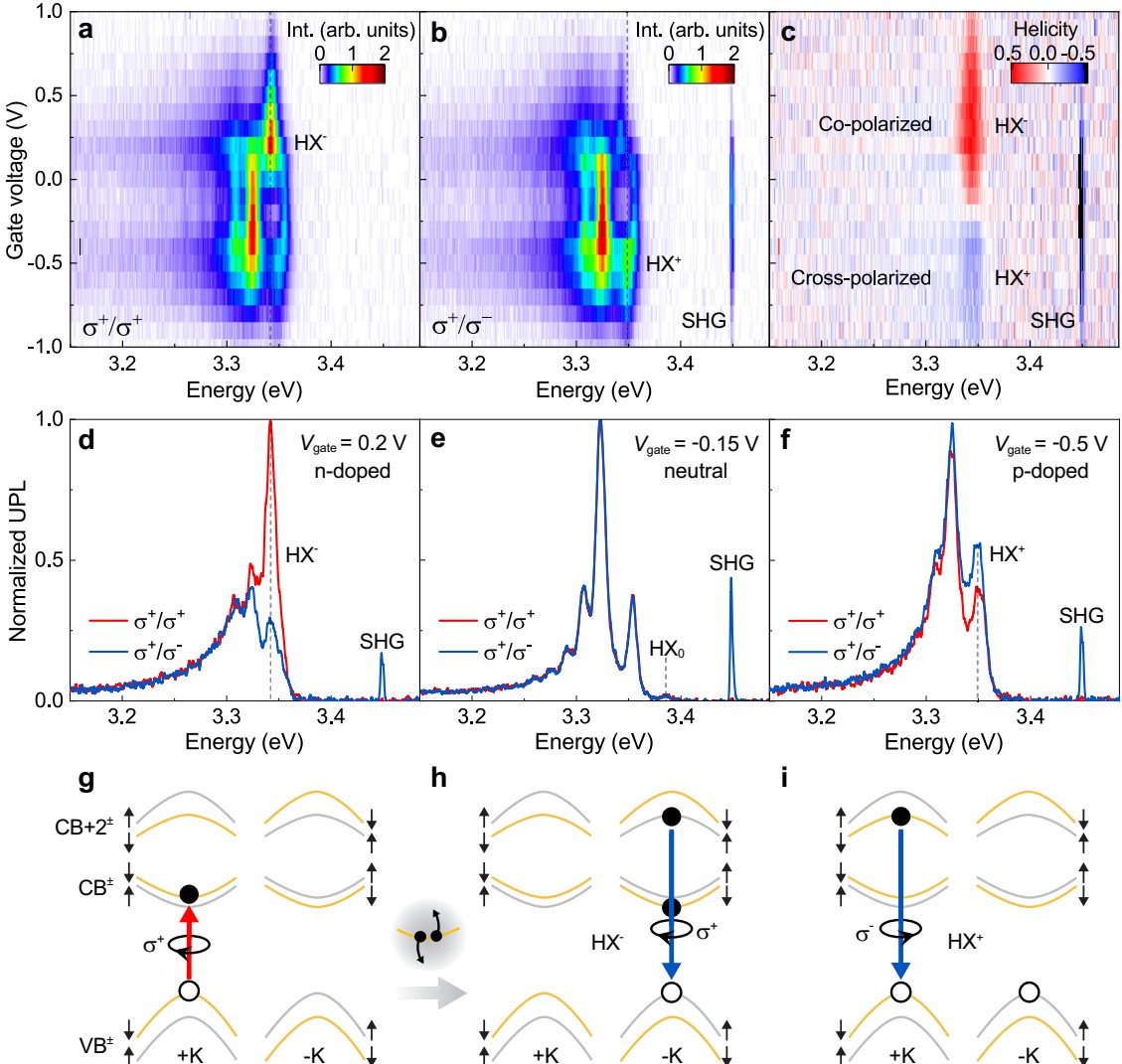

**Fig. 2 | Valley polarization and helicity of the high-lying trion in UPL from monolayer WSe₂.** $a$, $b$ Gate-voltage dependence of co-polarized ($a$) and cross-polarized ($b$) UPL under left-hand circularly polarized excitation. $c$ Helicity of the UPL as a function of gate voltage. $d$–$f$ Polarization resolved UPL spectra at gate voltages of 0.2 V ($d$), −0.15 V ($e$), and −0.5 V ($f$). $g$ Illustration of resonant pumping of the A exciton in the +K valley with a σ⁺ circularly polarized CW laser. This excitation

selectively polarizes resident spin-down electrons in the conduction band, CB. The inset illustrates the Auger-like exciton-exciton annihilation process that promotes the electron to high-lying conduction bands. $h$ Schematic of negatively charged high-lying trion in the −K valley with σ⁺ polarized emission and a resident spin-down electron in the −K valley. $i$ Schematic of the positively charged high-lying trion in the +K valley with σ⁻ polarized emission.

charged one, analogous to the situation for the band-edge trions (Fig. 1d)[27]. However, we are not able to resolve singlet and triplet HX⁻ species. These species may be masked by the phonon progressions, or are limited in population due to the formation process of the high-lying trions. As we discuss below, the high-lying trions can form through the capture of the resident charge carrier by the HX, which in turn was created in the course of either an Auger-like annihilation of two excitons in the ground state, or via an Auger-like annihilation process between a band-edge trion and a band-edge exciton. It is conceivable that the latter case shows a preference for certain internal structures of negative high-lying trions, as explained in Supplementary Note 2 and Supplementary Figs. 5–7.

The neutral HX PL spectrum is characterized by a distinct phonon progression, as seen in Fig. 1f at a gate voltage of −0.15 V, since the electron in the high-lying conduction band near the ±K-points couples strongly to zone-edge longitudinal acoustic (LA) phonons[16]. The peak intensities of the phonon replicas alternate with peak order number, indicating a transformation between momentum-direct, i.e., optically active, and momentum-indirect, i.e., optically inactive, states[28]. The

spacing of 31 meV between two bright peaks corresponds to the energy of two LA phonons, scattering the electron between valleys and back again[16]. The same spacing is resolved in the trion UPL spectra as marked by the curved black arrows in Fig. 1e. As shown in Supplementary Fig. 3, this phonon progression appears more pronounced after subtracting the UPL spectrum at charge neutrality (−0.15 V), confirming the excitonic origin of these peaks. Nevertheless, the zero-phonon lines for the charged HXs appear much stronger than that for the neutral HX, indicating a weaker electron-phonon coupling for the charged HXs.

## Valley polarization of high-lying trions

Next, we explore the valley polarization characteristics[29] of these charged HX by examining the degree of circular polarization (CP) in the UPL. Figure 2a, b shows the circular dichroism of the UPL as a function of gate voltage, with the corresponding helicity of the UPL plotted in panel C. The helicity is calculated by $(I_{\sigma^+/\sigma^+} - I_{\sigma^+/\sigma^-})/(I_{\sigma^+/\sigma^+} + I_{\sigma^+/\sigma^-})$, where $I_{\sigma^+/\sigma^+}$ ($I_{\sigma^+/\sigma^-}$) is the intensity of the co-polarized (cross-polarized) UPL. Figure 2d–f plots representative CP-resolved UPL spectra in

the electron-doping, neutral, and hole-doping regimes. The highest-energy peak in the spectra again arises from the CW SHG, which, in accordance with the selection rules intrinsic to the $C_3$ symmetry[30,31], always retains a CP opposite to that of the incident laser and therefore serves as a reference. Remarkably, the UPL from both HX trions is circularly polarized by up to 50%, whereas the UPL of the neutral HX appears unpolarized. This stark difference in the valley polarization of neutral and charged HXs could be due to the efficient electron-hole exchange interaction[32–34], the underlying formation mechanism as illustrated in Supplementary Fig. 5, or differences in their lifetime. Given that spin-valley locking is only effective in a limited region of momentum space around the K-points for the high-lying conduction band[16], the observed valley polarization of high-lying trions also corroborates our earlier finding that stable high-lying excitons must originate from the ±K-points[16].

Surprisingly, HX⁻ and HX⁺ exhibit opposite valley polarizations, and the CP UPL hence shows opposite helicity, in stark contrast to the band-edge trions, which generally exhibit the same helicity[1,27,29]. Helicity-resolved two-photon PL of the neutral and charged HX displays identical behavior as shown in Supplementary Fig. 4. The HX⁺ shows CP opposite to that of the excitation (Fig. 2g, i). This inversion agrees with the significant difference in selection rules for VB ↔ CB and VB ↔ CB+2 transitions in the same ±K valley[35,36], as summarized in Table 1. In contrast, the HX⁻ state is co-polarized with the laser, which indicates a localization at the −K valley according to the selection rule as sketched in Fig. 2h. We note that such an opposite helicity of positive and negative trions has recently been observed for band-edge excitons in monolayer WSe₂ under CW excitation and was rationalized by the resident-carrier polarization effect[37], i.e., the creation of a large polarization of resident electrons at the valley opposite to that of excitation. As illustrated in Fig. 2h and elaborated on in Supplementary Fig. 6, we therefore conclude that the most probable configuration of the HX⁻ is intravalley in nature. Because of the large spin−orbit splitting in the top valence band, this polarization process is not expected to be applicable to resident holes.

## High-lying excitons and trions in monolayer MoSe₂

Since the resident-carrier polarization effect relies on a fast intervalley scattering of the excited electron from the upper spin-split CB to the lower spin-split CB, it is not expected to arise in MoSe₂ monolayers, where photoexcited electrons reside in the lower spin-split CB. We would therefore expect positively and negatively charged high-lying trions in MoSe₂ to both have the same helicity, opposite to the helicity of the band-edge excitation. To test this hypothesis, we investigate the high-lying exciton and trions in a monolayer MoSe₂ transistor device.

Figure 3a presents the PL of band-edge excitons from monolayer MoSe₂ as a function of the gate voltage. We probe the HX by resonantly pumping the A exciton and measuring the high-energy UPL. Figure 3b shows the UPL and CW SHG of monolayer MoSe₂ as a function of the gate voltage. In stark contrast to the broad linewidth of the well-known C exciton[38,39], a narrow-band high-energy emission feature indeed appears right below the CW SHG, reminiscent of the HX in monolayer WSe₂. Figure 3c shows a rescaled plot, where the gate voltage dependence of this feature is clearly resolved and closely correlates with the

gate voltage dependence of the band-edge excitons in Fig. 3a. We assign this feature to the HX of monolayer MoSe₂ and the associated trions. Following our initial report of the HX in monolayer WSe₂[16], this observation constitutes the first experimental confirmation of the high-energy and narrow-linewidth state in another type of TMDC monolayer. The fact that HX⁺ and HX⁻ have the same binding energy in this case, in contrast to the case of WSe₂, matches well with the finding that positively and negatively charged A excitons have the same binding energy as seen in Fig. 3a and previously reported in ref. 5. This observation has been supported by the fact that the band-edge electrons and holes having the same effective mass[5]. We further characterize the helicities in the n-doped, neutral, and p-doped regimes. As shown in Fig. 3d–f, HX⁻ and HX⁺ have the same helicity, opposite to that of the excitation laser, which matches well with what is expected from the selection rules in Table 1 when there is no resident-carrier polarization effect. Supplementary Fig. 8 shows the SHG on the full scale.

## Theoretical considerations of high-lying trions

Next, we consider the high-lying trions with negative-mass electrons from a theoretical perspective. We first analyze the HX binding energies in the parabolic approximation for the bands. We introduce the hole effective mass, $m_h > 0$, the CB electron effective mass, $m_1 > 0$, and the effective mass of the high-lying electron from the CB+2 band, $m_2 < 0$. Corresponding electron-hole reduced masses are denoted as $\mu_1 = m_1 m_h/(m_1 + m_h)$ and $\mu_2 = m_2 m_h/(m_2 + m_h)$. Accordingly, $\mu_2 > 0$ because $|m_2| > m_h$[16]. In the case of HX⁺, the calculation of the trion binding energy can be performed following ref. 40 (see also refs. 41,42). This calculation yields the binding energy of the HX⁺ of approximately 10% of the HX binding energy, depending on the screening parameters and the effective masses, in reasonable agreement with experimental observations for monolayer WSe₂ summarized in Table 2 and Supplementary Table 1.

The situation with the HX⁻ is more involved. Due to the fact that the trion envelope function should be symmetric with respect to the permutation of identical charge carriers, while the antisymmetry of the total wavefunction results from the Bloch amplitudes, the problem for the HX⁻ is mapped, in the parabolic approximation, to the problem of the trion with an effectively reduced mass $\bar{\mu}^{-1} = (\mu_1^{-1} + \mu_2^{-1})/2$ and an effective-mass ratio $\sigma = 2m_1 m_2/[m_h(m_1 + m_2)]$. As detailed in Supplementary Note 3, the calculations show that, neglecting the non-local dielectric screening effects, the trion binding energy can be recast as

$$E_{b,HX^-} = \frac{2\mu_2 e^4}{\hbar^2 k^2}\left[\frac{\bar{\mu}}{\mu_2}(1+\chi) - 1\right].$$

Here, $\chi \equiv \chi(\sigma) \approx 0.1 - 0.5$ is the ratio of the trion to exciton binding energies in the case of equal effective masses of identical electrons. The analysis shows that for the bound trion to exist, $E_{b,HX^-} > 0$, the effective masses need to satisfy stringent conditions, $m_h < 2\chi m_1$ and $m_2 < m_1 m_h(1 + 2\chi)/(m_h - 2\chi m_1)$, which are not necessarily given for realistic band-structure parameters (see Supplementary Note 3 for a detailed discussion). Atomistic calculations show, however, that the dispersion of the high-lying CB+2 band is strongly non-parabolic. We analyze the role of quartic terms in the CB+2 dispersion taken in the simplest form $E_{CB+2} = \hbar^2 k^2/2m_2 + Bk^4$, with $B > 0$. Variational calculations presented in Supplementary Note 3 demonstrate that for $m_2 < 0$ and a not too small $B > 0$, bound HX⁻ states can exist with binding energies in the range of 0.1–1 of the exciton binding energy. We also confirm these variational calculations with the model of contact interaction of the exciton and free electron, accounting for non-parabolic terms in the dispersion. Consideration of dielectric screening by the Rytova-Keldysh potential[43] does not qualitatively change the results. However, a detailed comparison between the

**Table 1 | The selection rules of the group $C_{3h}$ at +K and −K valleys for circularly polarized PL**

| Transition | +K valley | | −K valley | |
|---|---|---|---|---|
| | Irrep | Helicity | Irrep | Helicity |
| VB ↔ CB+2 | $A' \leftrightarrow E_2'$ | $\sigma^-$ | $A' \leftrightarrow E_1'$ | $\sigma^+$ |
| VB ↔ CB | $A' \leftrightarrow E_1'$ | $\sigma^+$ | $A' \leftrightarrow E_2'$ | $\sigma^-$ |

*Irrep* irreducible representation.

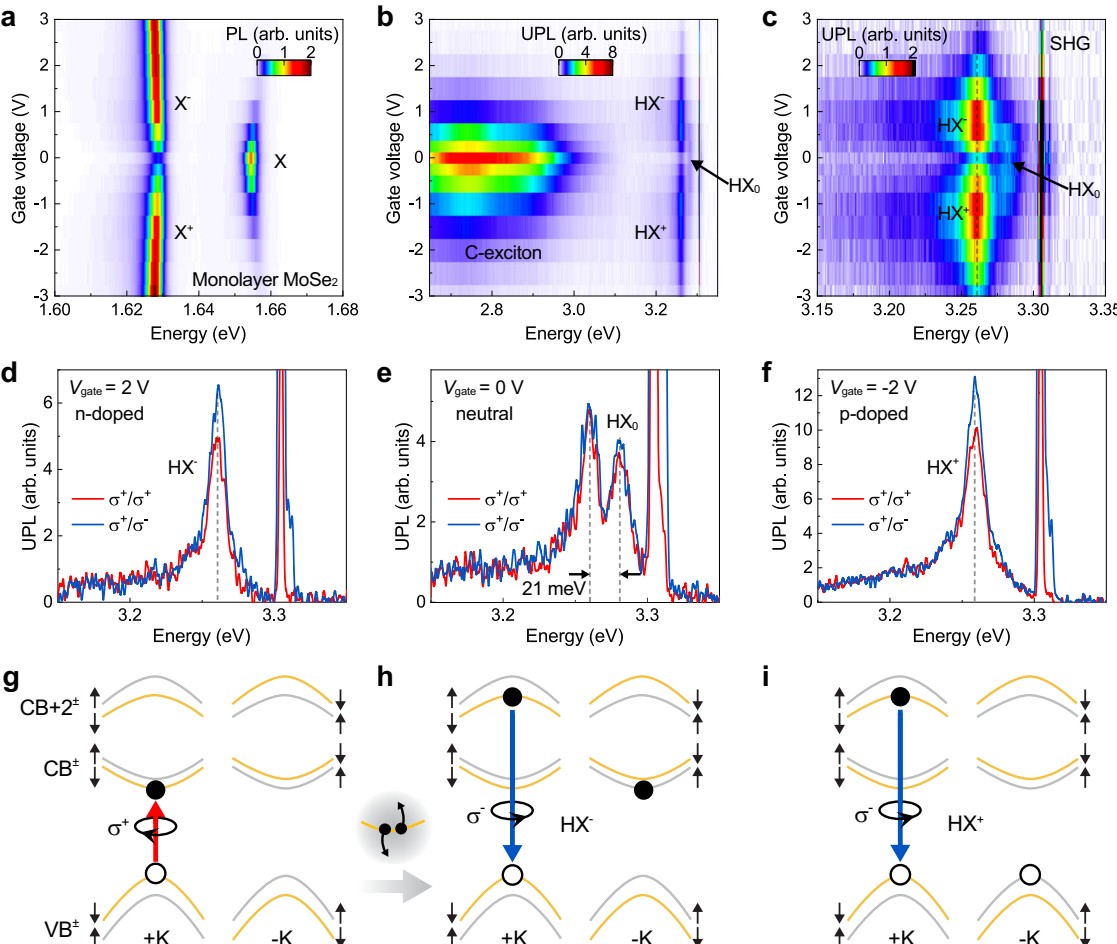

**Fig. 3 | High-lying excitons and trions in monolayer MoSe₂. a** Doping dependence of band-edge excitons (X) from monolayer MoSe₂, measured by the PL as a function of the gate voltage applied to a top graphite electrode. **b** UPL as a function of gate voltage. The excitation is at 1.654 eV in resonance with X. The features attributed to the neutral HX, and the negatively and positively charged HX are marked as HX₀, HX⁻, and HX⁺, respectively. **c** A rescaled plot of panel B highlighting the HX emission. **d–f** Polarization resolved

UPL spectra at gate voltages of 2 V (**d**), 0 V (**e**), and −2 V (**f**). **g** Illustration of resonant pumping scheme of the A exciton in the +K valley with a σ⁺ circularly polarized CW laser. The inset illustrates the Auger-like exciton-exciton annihilation process. **h** Schematic of the dominant negatively charged high-lying trion at the +K valley, with σ⁻ polarized emission. **i** Schematic of the dominant positively charged high-lying trion at the +K valley, with σ⁻ polarized emission.

---

experiment and theory, and fitting of the binding energies, requires accounting for the dielectric screening and the full dispersion of CB +2, including its anisotropy, and goes beyond the current work.

## Electrical control of excitonic quantum interference
Having presented experimental evidence and a theoretical rationalization of high-lying trions, we now turn to the influence of the electrical tunability of these species on excitonic quantum interference. Optical re-excitation of one and the same electron by a femtosecond laser pulse can drive direct transitions between the band edge and the high-lying conduction band, providing an optical coupling mechanism

## Table 2 | Experimental binding energies of the high-lying trions and band-edge trions in hBN encapsulated monolayer WSe₂ and MoSe₂

| High-lying trions | WSe₂ | | MoSe₂ | |
|---|---|---|---|---|
| | **HX⁻** | **HX⁺** | **HX⁻** | **HX⁺** |
| Binding energy (meV) | 43 | 35 | 21 | 21 |
| **Band-edge trions** | **X$_S^-$** | **X$_T^-$** | **X⁻** | **X⁺** |
| Binding energy (meV) | 36 | 29 | 21 | 26 | 26 |

that interconverts the band-edge A exciton and an HX state[16,17]. Such a high-energy state is clearly identified in the two-photon excitation spectrum but not in the luminescence[16], and is therefore attributed to a dark *p*-like HX. Because of the lack of an inversion center in monolayer WSe₂, both the *s*-like and *p*-like excitons at the band edge are mixed and can be simultaneously one- and two-photon active[44]. However, the *p*-states are expected to dominate the two-photon absorption[44]. The same should be true for the trions. Together with the ground state of the system, i.e. the state where no exciton is present, an excitonic three-level system is formed. Interactions with the light field can then be treated in analogy to the familiar case of atomic multi-level systems in quantum optics[17,45]. As illustrated in the left panel of Fig. 4b, laser-driven transitions between states can undergo Rabi oscillations. The associated quantum interference between |1⟩ → |2⟩ and |1⟩ → |2⟩ → |3⟩ → |2⟩ transition pathways is then observed in the SHG spectrum generated by a femtosecond laser pulse[17]. These interferences appear as dips in the SHG spectrum along with a characteristic spectral anti-crossing feature in the excitation-energy dependence of the SHG spectra, shown in the left panels of Fig. 4a.

In analogy to the neutral excitonic three-level system reported previously[16,17], it is conceivable that a band-edge trion and a *p*-like HX trion can also form a three-level system, which should give rise to an

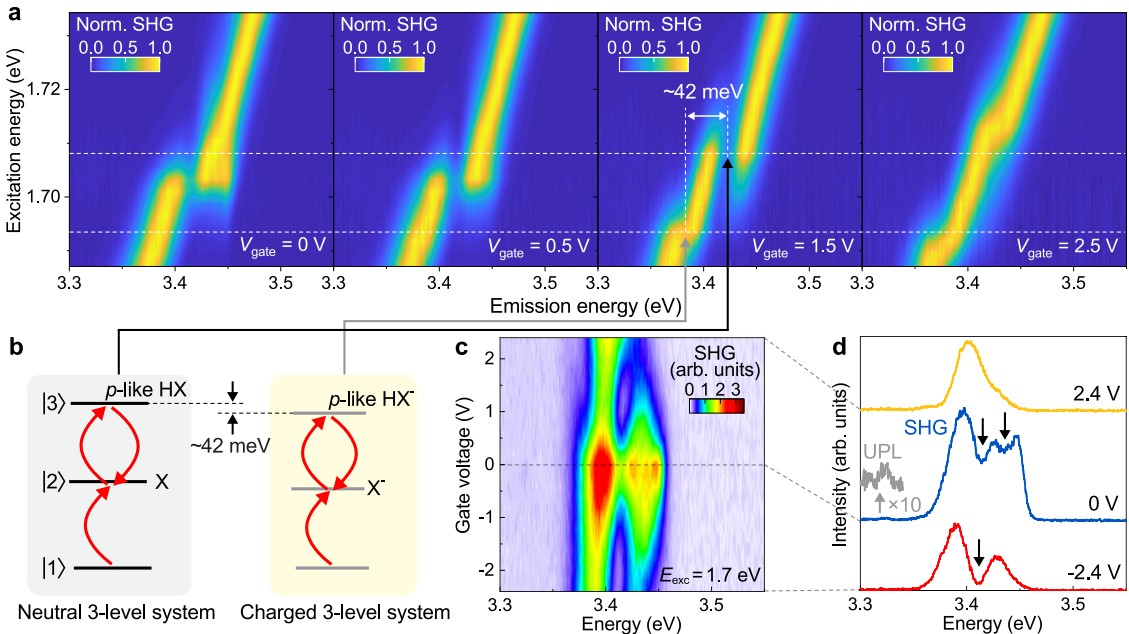

**Fig. 4 | Electrical control of excitonic quantum interference involving the _p_-like high-lying trion in monolayer WSe₂. a** Excitation-energy dependence of SHG spectra at gate voltages of 0, 0.5, 1.5, and 2.5 V, measured under excitation by a laser of 80 fs pulse length. **b** Illustration of an excitonic three-level system based on neutral and charged species. The neutral three-level system is formed by the band-edge A exciton and the _p_-like HX. A negatively charged three-level system is formed by the band-edge trion and the charged _p_-like HX. **c** Gate-voltage dependence of the SHG spectrum measured at an excitation energy of 1.7 eV. **d** SHG spectra at gate voltages of 2.4, 0, and −2.4 V. Black arrows mark the SHG dips. The gray line presents the same spectrum at 0 V (blue line) multiplied by 10 at the corresponding energy range, showing the UPL of the _s_-like HX. All measurements were performed at 5 K.

additional quantum-interference feature in the resonant SHG of the coherently driven system. We note that the stability of excited trion states is a separate theoretical problem, see refs. 26,40,46 for details, and thus refrain here from a precise assignment of the excited charged HX state involved in the quantum interference process. We use the term "charged _p_-like HX" here for brevity.

Figure 4a shows the gate-voltage dependence of SHG excitation spectra from monolayer WSe₂. The measurement was carried out with a wavelength-tunable laser with 80 fs pulse duration. With increasing gate voltage and thus electron-doping density, the A exciton energy and the corresponding anti-crossing feature in the spectrum shift to higher energies. Remarkably, an additional anti-crossing feature (gray arrow in the second panel from the right) appears at a gate voltage of 1.5 V, 42 meV below the main transparency dip. For ladder-type three-level systems, the position of this dip translates directly to the energy of the associated high-lying state[17,47], which in this case can tentatively be assigned to the _p_-like HX⁻ trion. Interestingly, upon further increase of the gate voltage, this additional anti-crossing feature shifts to the red, opposite to the shift direction of the main anti-crossing feature. This opposite doping dependence matches well with the expectations for a charged and a neutral exciton, as outlined in Supplementary Fig. 1d for the band-edge A exciton and the corresponding trions. We therefore assign this additional spectral anti-crossing feature to the charged _p_-like HX and obtain an energy difference of roughly 42 meV between the charged and the neutral _p_-like HX. This energy difference, measured by the resonant SHG, coincides perfectly with the 43 meV energy difference between the _s_-like HX⁻ and the zero-phonon line of the _s_-like HX found in the UPL spectra in Fig. 1f.

Signatures of quantum interference in the SHG spectrum also provide insight into the coherence time of the excitonic species, which is a crucial parameter enabling the quantum-interference phenomenon as shown by simulations in the density-matrix formalism[17]. Since similar spectral features are observed in the neutral and charged excitonic three-level system, we conclude that the coherence times of the excitons and the trions must be comparable. This conclusion of a

significant coherence time of the trion is supported by the relatively weak effect of the electron-trion scattering, consistent with previous work on trions and the Fermi polaron description of the effect[6,26,48,49].

Finally, we evaluate the potential to control the excitonic quantum interference electrically. The electrical gate tunes the charge-carrier density in the monolayer and leads to a change in both the oscillator strength and the detuning of the excitonic transitions (Supplementary Fig. 1d). These changes are expected to alter the Rabi frequency. Figure 4c shows the voltage dependence of the SHG spectrum generated under 1.7 eV excitation, revealing a dramatic change in the spectral structure. Figure 4d exhibits three examples of SHG spectra at gate voltages of 2.4, 0, and −2.4 V. With simulations of the density-matrix dynamics it can be shown that each dip in the SHG spectrum corresponds to one full Rabi cycle of the strongly driven system[17]. At 2.4 V, the SHG spectrum does not show any prominent dip: in this case, no Rabi flopping occurs. At −2.4 V, one clear dip emerges in the spectrum, implying one Rabi cycle. At 0 V, two dips are identified, implying that the driven system must undergo two Rabi cycles. Such an evolution of the number of Rabi cycles with electrical gate voltage demonstrates an unprecedented control over a quantum-optical phenomenon in the form of excitonic quantum interference.

## Discussion

We have demonstrated the existence of UV-emissive trions in both WSe₂ and MoSe₂ monolayer transistor structures. These unusual excitonic species exhibit a high degree of valley polarization, corroborating our earlier conclusion that high-lying excitons originate from the ±K-points in momentum space and consist of an electron from a high-lying conduction band CB+2[16]. We systematically studied the high-lying trions comprising negative-mass electrons in theory and have uncovered a broad set of conditions under which such trions can be stable. In addition to the bright high-lying trions, we identify a dark high-lying trion that couples with band-edge trions to form a charged excitonic three-level system, which enables laser-driven excitonic quantum interference. We show that excitonic quantum interference

from both neutral and charged excitonic three-level systems can be controlled by the gate voltage. The number of Rabi cycles undergone during the laser pulse can be tuned via the gate voltage without changing the laser power. Such electrical control of quantum interference is not conceivable in conventional quantum-optical experiments on dilute atomic gases. Our findings therefore expand the spectral working range of future valleytronic devices to the UV, and open up new possibilities for quantum nonlinear optoelectronics.

## Methods

### Device fabrication

We fabricate the monolayer $WSe_2$ and $MoSe_2$ transistors by a dry-transfer method[50]. Monolayer $WSe_2$, monolayer $MoSe_2$, few-layer hexagonal boron nitride, and few-layer graphite flakes are exfoliated from bulk crystals ($WSe_2$ and $MoSe_2$, HQ Graphene; hBN, NIMS) onto PDMS films (Gel-Pak, Gel-film X4) using Nitto tape. We stack these layers onto a $Si/SiO_2$ substrate with prepatterned gold electrodes using a stamping method. A microscope image of a representative device is shown in Fig. 1c.

### Optical spectroscopy

We cool down the sample to 5 K in a helium-flow microscope cryostat (Janis, ST-500). We focus the laser onto the sample with an objective of 0.6 numerical aperture (Olympus, LUCPLFLN, 40×) and measure reflected signals. To measure the PL, we use an argon-ion laser (Spectra Physics, 2045E) at 488 nm for excitation and a 488 nm long-pass edge filter to remove the laser line. To measure the UPL, we use a tunable continuous-wave laser (Sirah, Matisse CR) for excitation and a 680 nm short-pass filter to remove the laser line. For the helicity-resolved measurements, we use a Berek compensator (Newport) to generate the circularly polarized excitation and determine the signal polarization through a combination of a superachromatic quarter-wave plate and a polarizer. To measure SHG, we use a tunable pulsed Ti:sapphire laser (Mai Tai XF, 80 MHz repetition rate) for excitation and a 680 nm short-pass filter to remove the laser line. For both PL and UPL measurements, we use a grating of 600 grooves $mm^{-1}$ to disperse the signals and a CCD camera (Princeton Instruments, PIXIS 100) for detection. For SHG, a grating of 1200 grooves $mm^{-1}$ is used.

## Data availability

Source data for figures are provided with the paper. Any additional data are available from the corresponding authors upon reasonable request.

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

## Acknowledgements
The authors thank Paulo Eduardo Faria Junior and Yaroslav Zhumagulov from the group of Jaroslav Fabian for many helpful discussions, Christian Bäuml and Nicola Paradiso from the group of Christoph Strunk for preparing substrates with prepatterned gold electrodes, and Sebastian Krug for technical support. Financial support is gratefully acknowledged from the Deutsche Forschungsgemeinschaft (DFG, German Research Foundation) through SFB 1277 (Project-ID: 314695032) projects B03, B05, and B11, SPP 2244 (Project-ID: LI 3725/1-1, 443378379), an Emmy-Noether Grant (Project-ID: CH 1672/1-1, 287022282), and the Würzburg-Dresden Cluster of Excellence on Complexity and Topology in Quantum Matter ct.qmat (EXC 2147, Project-ID: 390858490). Growth of hexagonal boron nitride crystals was supported by the Elemental Strategy Initiative conducted by the MEXT, Japan (Grant Number JPMXP0112101001) and JSPS KAKENHI (Grant Numbers 19H05790 and JP20H00354). M.M.G. is grateful to the Russian Science Foundation, Grant No. 19-12-00051 (analytical theory); M.A.S. acknowledges support from RFBR Grant No. 19-52-12038 (numerical computations).

## Author contributions
K.-Q.L. conceived the project and carried out the measurements with the support of S.B.; J.D.Z. and A.C. fabricated the gate-tunable devices; M.M.G., M.A.S., and J.V.M. contributed the theoretical work; K.W. and T.T. provided hBN crystals. K.-Q.L., J.D.Z., M.A.S., J.V.M, S.B., A.C., M.M.G., and J.M.L. discussed the results, analyzed the data, and contributed to the writing of the manuscript.

## Funding

## Competing interests
The authors declare no competing interests.
