## [Peer Review File · Nature Communications]

High-lying valley-polarized trions in 2D semiconductorsReviewers' Comments:

Reviewer #1:

Remarks to the Author:

The authors studied high-lying excitonic states in WSe₂ and MoSe₂, including the electron from upper conduction band with negative effective mass. They reported and explained the distinct optical selection rules for high-lying trions. The authors also identified a dark high-lying trion which couples with band-edge trion to form a charged excitonic three-level system and showed that the gate voltage can control this three-level system.

Although the topic is timely, the experimental results and explanation are interesting, the theory part is still weak. Firstly the author used the potential $V(q) \sim 1/q$, instead of the Keldysh potential $V_K(q) \sim 1/q(1+r_0 q)$, to calculate the binding energy. Please note that the $1/q$ potential is a good approximation only in long range limit $q \rightarrow 0$ whereas the trion binding energy is gained from the short-range interaction (dipole-charge interaction). The $1/q$ is therefore a bad approximation for this case. Secondly, using the trial function with only several parameters doesn't guarantee that the obtained wave function is the optimal one. Lastly, the theory cannot provide a direct comparison with the experimental data although the effective mass of the upper conduction band has been given in Ref. 16. Above points makes me not recommend the manuscript for a publication in Nature Communications, at least for the current version.

Below are several points/questions which might help to improve the manuscript.

1. The right panel of figure 1a is confusing. The reader might think that the two holes belong to the same valley. The authors should modify the figure to avoid this confusion.
2. Please check the number of 734 nm for the wave length of excitation light in the caption of Fig S3. The energy of such light is 1.6892 eV. How can its SHG be at higher than 3.4 eV?
3. Why are zero-phonon lines of HX^{+} and HX^{-} strong whereas one for HX^0 is much weaker ?
4. What is the relative intensities between the high-lying excitonic peaks and band-edge peaks?

Reviewer #2:

Remarks to the Author:

The authors report experimental evidence of high-lying trions in a monolayer WSe₂ transistor and MoSe₂ transistor. Those high-lying trions have different optical selection rules compared to band-edge trions and show helicity opposite to that of the excitation. A dark charged p-like HX is identified by the signature of excitonic quantum interference in SHG. A dark high-lying trion that couples with band-edge trions to form a charged excitonic three-level system, enables laser-driven excitonic quantum interference. This work is interesting to the 2D material community as well as to a broad audience, expanding the spectral working range of future valleytronic devices to the UV. The paper is well written, original, and scientifically sound. However, if the authors can provide more experimental result and explanation, it will be more convincing to readers. I encourage the authors to add more discussions based on referee comments.

1. In Fig. 1a, for the positively charged HX, it is not clear if the photoexcited hole and the additional resident hole is at the same K-point or not. The label in figure 1a, "+K/-K", is not clear. It is not clear whether the additional resident hole/electron is at the same valley of the photoexcited electron-hole pair. A modification of the figure 1a or more detailed description of the charged HX is recommended.
2. In line 144, the difference in the valley polarization of neutral and charge HXs could be due to the efficient electron-hole exchange interaction or differences in their lifetime. Is there any time resolved PL data available to extract the lifetime of the neutral and charge HXs? The lifetime information is critical to understand the dynamic and origin of the difference in the valley polarization.

3. The author mentioned the inset illustrates the Auger-like exciton-exciton annihilation process that promotes the electron to high-lying conduction bands. The exciton-exciton annihilation process should be a nonlinear process and shows a super linear relation with the density of exciton. A dependence of laser power vs the intensity of UPL will be strong evidence of the Auger-like process.
4. Could the author list the binding energy of HX^+ , HX^- , X^+ and X^- explicitly in the main text? It will make a stronger justification for line 202.
5. In line 274, interferences appear as dips in the SHG spectrum along with a characteristic spectral anti-crossing feature in the excitation-energy dependence of the SHG spectra. In the data analysis of the figure 4d, should the SHG signal be normalized with the original laser spectrum in order to find the energy of dip?
6. In fig 3d e f, the laser SHG is not perfectly cancelled in the co-polarized configuration. The SHG intensity ratio of the two configurations is not shown clearly in fig 3d e f. I suggest adding a color plot of Helicity of $MoSe_2$, similar as fig 2c into fig3 or supplementary materials.
7. During the data analysis of fig4, the existence of the dip and the dip position of the SHG signal is critical to understand the quantum-interference feature in the resonant SHG of the coherently driven system. While in the fig 4c, the 1.7eV laser is very close to the exciton resonance frequency. In the color plot of fig 4c, will UPL and SHG contribute together to the spectra? If UPL also contributes to the spectra, the data analysis of the SHG will be problematic and the existence of the dip and the dip position will be not convincing.

We cordially thank the referees for their thorough review of our work and their insightful queries, which helped us to strengthen our manuscript. Naturally, we are encouraged to receive positive assessments – “*the topic is timely, the experimental results and explanation are interesting, ...*” (#1), “*This work is interesting to the 2D material community as well as to a broad audience, expanding the spectral working range of future valleytronic devices to the UV.*” (#2).

We recognize that the referees have some concerns and comments about the paper, based on which we have now improved the work extensively. In the following, we provide a point-by-point response to the comments. The changes made to the manuscript and SI are also indicated in the marked-up versions attached to this document.

RESPONSE TO REVIEWER COMMENTS

Reviewer #1 (Remarks to the Author):

The authors studied high-lying excitonic states in WSe₂ and MoSe₂, including the electron from upper conduction band with negative effective mass. They reported and explained the distinct optical selection rules for high-lying trions. The authors also identified a dark high-lying trion which couples with band-edge trion to form a charged excitonic three-level system and showed that the gate voltage can control this three-level system.

Although the topic is timely, the experimental results and explanation are interesting, the theory part is still weak. Firstly the author used the potential $V(q) \sim 1/q$, instead of the Keldysh potential $V_K(q) \sim 1/q(1+r_0 q)$, to calculate the binding energy. Please note that the $1/q$ potential is a good approximation only in long range limit $q \rightarrow 0$ whereas the trion binding energy is gained from the short-range interaction (dipole-charge interaction). The $1/q$ is therefore a bad approximation for this case. Secondly, using the trial function with only several parameters doesn't guarantee that the obtained wave function is the optimal one. Lastly, the theory cannot provide a direct comparison with the experimental data although the effective mass of the upper conduction band has been given in Ref. 16. Above points makes me not recommend the manuscript for a publication in Nature Communications, at least for the current version.

Response 1.0: We thank the referee for his/her time to carefully examine our manuscript, for agreeing with us that it is an interesting, peculiar effect, and for the instructive comments, which we address below in detail.

We appreciate the important points raised by the referee regarding the theoretical part of the work. We have taken the criticism seriously and performed additional calculations and analysis. The response is structured in three parts below, corresponding to the three main points from the referee.

1. First, the referee emphasizes the importance of the chosen potential. Indeed, for simplicity and clarity of presentation we have chosen the Coulomb potential instead of the Rytova-Keldysh potential. We agree with the referee that the Rytova-Keldysh potential provides a better approximation of the electron-hole interaction in two-dimensional transition-metal dichalcogenides. We have therefore now performed preliminary additional calculations using the screened Rytova-Keldysh potential. In general, these calculations lead to similar values of both the trion and exciton binding energies, but there is a tendency for a somewhat smaller ratio of the trion to exciton binding energy in the case of screening. For instance, with the screening parameter $r_0 = a_B = 1.5 \text{ nm}$,

reduced masses $\mu_1 = \mu_2$, and a quartic term in the dispersion $B^* = B \frac{\mu_1^3 e^4}{\kappa^2 \hbar^6} = 1$, we find $\frac{E_{B,HT}}{E_{B,HX}} \approx 0.18$ as compared to the ratio of 0.32 for the case of a purely Coulombic potential. We agree that a detailed theoretical analysis of the effect of the form of the potential on the high-lying trion binding energy deserves further studies, but appears to be beyond the scope of the current work. However, the calculations using the Coulomb potential therefore serve as a first, reasonable approximation supporting the experimental observations, which are the focus of the present paper.

2. The second point relates to the applicability of the variational approach and the corresponding trial functions for the trions. Our response is twofold:

a) Corresponding trial functions for excitons and trions with positive reduced masses have already been tested for a large variety of two-dimensional and quasi-two-dimensional structures, including conventional quantum wells^{R1}, transition-metal dichalcogenide monolayers^{R2-R4} and bilayers^{R5}. In particular, we have previously compared the variational approach with more advanced calculation techniques and have demonstrated a high degree of accuracy (with a deviation of less than 10% - 15% in typical cases) of this method in Ref. R4.

R1. R. A. Sergeev and R. A. Suris, Ground-state energy of X^- and X^+ trions in a two-dimensional quantum well at an arbitrary mass ratio, *Phys. Solid State* 43, 746 (2001)

R2. T. C. Berkelbach, M. S. Hybertsen, D. R. Reichman, Theory of neutral and charged excitons in monolayer transition metal dichalcogenides, *Phys. Rev. B* 88, 045318 (2013)

R3. M. Szyniszewski, E. Mostaani, N. D. Drummond, and V. I. Fal'ko, Binding energies of trions and biexcitons in two-dimensional semiconductors from diffusion quantum Monte Carlo calculations, *Phys. Rev. B* 95, 081301 (R) (2017)

R4. E. Courtade, M. A. Semina, M. Manca, M. M. Glazov et al., Charged excitons in monolayer WSe_2 : Experiment and theory, *Phys. Rev. B* 96, 085302 (2017)

R5. M. A. Semina, Excitons and trions in bilayer van der Waals heterostructures, *Phys. Solid State* 61, 2218 (2019)

b) The variational method provides an upper bound for the ground-state energy. We have now performed additional calculations of the HX energy using two different methods: (i) the variational method and (ii) a numerical method decomposing the wavefunctions over a large set of Gaussians (with the numerical convergence carefully checked). The results are presented in Fig. R1.0 below and both methods provide very close values of the exciton binding energy with the deviation being below $\sim 1\%$. The trion binding energy is determined by the difference between the total trion energy and the exciton energy. As we calculate the exciton energy to high accuracy, our variational calculation provides a lower bound for the trion binding energy. Our comparison with preliminary numerical calculations also shows that the error in the trion binding energy is actually small.

With this new evidence and the arguments made above, we are convinced that the accuracy of the variational method is sufficiently high to lay the theoretical foundations of the current work, reporting first experimental proof for the existence of novel high-lying trions. The additional numerical results including Fig. R1.0 are shown in the Supplementary Note 3-B.

Figure R1.0 | Exciton energy (i.e. the binding energy with a negative sign) as a function of the quartic term in the dispersion $B^* = B \frac{\mu_1^3 e^4}{\kappa^2 \hbar^6}$, calculated using the variational approach (symbols) and using a numerical diagonalization of the Hamiltonian (lines) for two ratios of the reduced masses. The Rytova-Keldysh potential with the screening parameter $r_0 = a_B = 1.5 \text{ nm}$ was used in the calculation. The variational approach matches perfectly with the numerical calculation.

3. The last critical point of the referee is the claim that the theory cannot deliver accurate values for the binding energies to compare directly with the experimental data. In fact, our initial goal was indeed to obtain quantitative theoretical values that we can compare to our experiment. However, even though we (our collaborators from Ref. 16, Chin Shen Ong, Diana Qiu and Steven Louie) succeeded in performing *ab initio* GW-BSE calculations of the high-lying exciton with 8 bands, the additional complexity of the high-lying trions renders a similar approach impossible at the moment. The methodology (*ab initio* GW-BSE for trions) is unfortunately not ready for this yet. Moreover, the role of dielectric environment of hBN would be missing in this approach as well. Effective model using the effective masses of the three bands (VB^+ , CB^- , and $CB+2^-$) under assumptions of parabolic dispersion unfortunately do not deliver a bound trion state.

Therefore, instead of aiming for accurate calculations of the binding energies, we have tried to develop a general qualitative understanding of the conditions under which a trion can form with an electron of negative effective mass. With the current approaches, we have demonstrated the plausibility of the high-lying trions in theory, presented the conditions under which they can be bound, analyzed the form of their wavefunctions, and estimated the bounds for their binding energies. From this perspective, the theory discussed in the manuscript is more general to different types of transition-metal dichalcogenide monolayers.

In summary, based on the above arguments and the additional calculations, we hope that the reviewer agrees with us that the current state of theoretical work should make a sufficiently strong case for rationalizing the remarkable experimental observations reported in this manuscript. Following reviewer's comment, we have included additional Supplementary Fig. 11 with the comparison of numerical and variational results (Fig. R1.0 above) and brief comments in the main text and supplement with clarifications on the choice of potential and accuracy of calculations.

Below are several points/questions which might help to improve the manuscript.

1. The right panel of figure 1a is confusing. The reader might think that the two holes belong to the same valley. The authors should modify the figure to avoid this confusion.

Response 1.1: We agree with the referee and have modified the figure to avoid this confusion now.

2. Please check the number of 734 nm for the wave length of excitation light in the caption of Fig S3. The energy of such light is 1.6892 eV. How can its SHG be at higher than 3.4 eV?

Response 1.2: We thank the referee for the accurate reading and pointing out this unfortunate error. It should be 726 nm, which gives rise to an SHG peak at 3.416 eV. We have corrected this and checked all of the numbers throughout the manuscript.

3. Why are zero-phonon lines of HX^+ and HX^- strong whereas one for HX^0 is much weaker?

Response 1.3: We consider that this is because the high-lying triions have a weaker coupling with the zone-edge longitudinal acoustic (LA) phonons than the neutral high-lying excitons. The phonon progression of the high-lying triions therefore has a much smaller Huang-Rhys factor. This conclusion is supported by the calculated coupling strengths of LA(K) phonons with different electronic states, shown in Fig. R1.1. Electrons in the $\text{CB}+2^-$ band couple much more strongly with the LA(K) phonons than electrons in the CB^- band or holes in the VB^+ band. This discussion was missing in the original manuscript, and we have now added it to the main text.

Figure R1.1 | Energy level correction of electronic states at the K-point with respect to the LA(K) phonon as a function of the amplitude of a specific “frozen” phonon, revealing much stronger electron-phonon coupling for electronic states in the $\text{CB}+2^-$ band than in the CB^+ and VB^+ bands. “ u ” represents the normal-mode amplitude of the LA(K) phonon. [Reproduced from Nat. Commun. 12, 5500 (2021)]

4. What is the relative intensities between the high-lying excitonic peaks and band-edge peaks?

Response 1.4: It is, unfortunately, not straightforward to directly compare the intensity of the UPL of high-lying excitons (HX) and the PL of the band-edge excitons (AX). The one-photon PL of the HX is masked by direct transitions from all over k -space, i.e. not only at the K-points, whereas UPL excites the K-point transitions selectively. However, we can compare the UPL intensity of the HX and the B exciton directly. As shown in Figure R1.2 below, the B exciton appears about 50 times brighter than the HX. Since the detection efficiency of our CCD camera is about 5 times lower in

the 3.4 eV region than in the 2.2 eV region, the UPL intensity of the B exciton is about 10 times stronger than that of the HX. The A exciton is about 100 times brighter than the B exciton and therefore about 1000 times brighter than the HX.

Figure R1.2 | a, Experimental reflectance contrast of monolayer WSe₂ encapsulated by hBN on a sapphire substrate at 5 K. **b**, UPL spectrum (solid line) measured under resonant pumping of the A-exciton. The dashed line shows the same spectrum multiplied by 0.02 in the energy range of the B-exciton. [Reproduced from Nat. Commun. 12, 5500 (2021)]

Reviewer #2 (Remarks to the Author):

The authors report experimental evidence of high-lying trions in a monolayer WSe₂ transistor and MoSe₂ transistor. Those high-lying trions have different optical selection rules compared to band-edge trions and show helicity opposite to that of the excitation. A dark charged p-like HX is identified by the signature of excitonic quantum interference in SHG. A dark high-lying trion that couples with band-edge trions to form a charged excitonic three-level system, enables laser-driven excitonic quantum interference. This work is interesting to the 2D material community as well as to a broad audience, expanding the spectral working range of future valleytronic devices to the UV. The paper is well written, original, and scientifically sound. However, if the authors can provide more experimental result and explanation, it will be more convincing to readers. I encourage the authors to add more discussions based on referee comments.

Response 2.0: We are grateful to the referee for his/her careful review of the manuscript, appreciation of our work, and extremely valuable comments that helped us to improve the manuscript. We revised the manuscript accordingly as detailed below.

1. In Fig. 1a, for the positively charged HX, it is not clear if the photoexcited hole and the additional resident hole is at the same K-point or not. The label in figure 1a, "+K/-K", is not clear. It is not clear whether the additional resident hole/electron is at the same valley of the photoexcited electron-hole pair. A modification of the figure 1a or more detailed description of the charged HX is recommended.

Response 2.1: We thank the referee for pointing out the confusion regarding the figure. We have corrected the Figure 1a and have also added a more detailed description of the charged HX to the figure caption.

2. In line 144, the difference in the valley polarization of neutral and charge HXs could be due to the efficient electron-hole exchange interaction or differences in their lifetime. Is there any time resolved PL data available to extract the lifetime of the neutral and charge HXs? The lifetime

information is critical to understand the dynamic and original of the difference in the valley polarization.

Response 2.2: We agree with the Referee that the valley polarization in the high-lying trion could well originate from the trion lifetime being much longer than that of the neutral high-lying excitons. We have tried different approaches, including time-correlated single photon counting (TCSPC) and synchroscan streak camera spectroscopy, to measure the lifetime of the neutral and charged HX. Unfortunately, these excitons can only be generated through the upconverted photoluminescence (UPL) process and are 1000 times dimmer than the A-exciton PL. Moreover, the UPL process convolutes the lifetime of the high-lying exciton with that of the band-edge excitons involved in the Auger-like exciton-exciton annihilation process. Although our preliminary evidence does suggest that the high-lying trions do have a longer lifetime than the neutral high-lying exciton, it is actually what would be expected. Whether this is the sole reason for the dramatically prolonged valley polarization, however, is another matter and cannot be resolved conclusively at the moment.

3. The author mentioned the inset illustrates the Auger-like exciton-exciton annihilation process that promotes the electron to high-lying conduction bands. The exciton-exciton annihilation process should be a nonlinear process and shows a super linear relation with the density of exciton. A dependence of laser power vs the intensity of UPL will be strong evidence of the auger-like process.

Response 2.3: We have added a power dependence of the UPL in Fig. R2.1 (new Supplementary Fig. 2). The UPL of the HX should indeed scale superlinearly with the exciton population density. However, because of a bleaching of the absorption band, the A-exciton density scales sublinearly with the pump power at the high pump fluences. Therefore, the UPL intensity of the HX transition is not necessarily quadratic in the pump power. Nevertheless, second-harmonic generation (SHG) generally scales parabolically with the excitation power and therefore serves as a suitable reference in the measurement when it is resonantly enhanced by the A-exciton transition. We plot both the continuous-wave SHG and the HX UPL intensity as a function of the pump power in Fig. R2.1. The UPL intensity of the HX shows a power-law exponent of 1.26 ± 0.04 , which is very close to that of 1.35 ± 0.04 found for the SHG intensity measured simultaneously (reduced from the canonical exponent of 2 because of the bleaching of the A-exciton transition), thus supporting the notion of an Auger-like exciton-exciton annihilation process being responsible for the HX UPL.

Figure R2.1 | Pump power dependence of upconverted photoluminescence (UPL) and continuous-wave second-harmonic generation (SHG). **a**, Simultaneous measurement of the UPL and SHG from monolayer WSe_2 encapsulated by hBN at 5 K as a function of excitation power of the narrow-band continuous-wave laser at 1.726 eV. **b**, Power dependence of the integrated HX

UPL and SHG intensity. The HX UPL shows a power law almost identical to that of the SHG, suggesting an Auger-like exciton-exciton annihilation process for the UPL, and both UPL and SHG are impacted by the bleaching of the A-exciton absorption at high fluences.

4. Could the author list the binding energy of HX^+ , HX^- , X^+ and X^- explicitly in the main text? It will make a stronger justification for line 202.

Response 2.4: We thank the referee for this instructive query. We have now added a new table (Table 2 in the main text) listing the binding energies of the high-lying trions and band-edge trions, which indeed make the story much easier to follow.

5. In line 274, interferences appear as dips in the SHG spectrum along with a characteristic spectral anti-crossing feature in the excitation-energy dependence of the SHG spectra. In the data analysis of the figure 4d, should the SHG signal be normalized with the original laser spectrum in order to find the energy of dip?

Response 2.5: Normalizing the SHG spectrum (recorded under femtosecond pulsed excitation rather than under continuous-wave excitation as in Fig. R2.1) to the original laser spectrum indeed appears to be an appealing approach for the analysis. However, it turns out that excitonic quantum interference not only causes dips in the SHG spectrum, but also shifts and broadens the spectrum. To determine the exact energy of state $|3\rangle$, we compare the experimental SHG spectrum with the simulated SHG spectrum using the density-matrix formalism. Most importantly, we find that the position of the dip does not shift significantly with the excitation energy, as seen in the anticrossing feature of the spectra in Fig. 4a (the 2D plots of excitation vs. emission energy), and in the raw data shown in Fig. R2.2 below.

Figure R2.2 | Dependence of the experimental SHG spectrum on the excitation energy E_{exc} , showing a negligible shift of the dip position with the excitation energy.

6. In fig 3d e f, the laser SHG is not perfectly cancelled in the co-polarized configuration. The SHG intensity ratio of the two configurations is not shown clearly in fig 3d e f. I suggest adding a color plot of Helicity of MoSe₂, similar as fig 2c into fig3 or supplementary materials.

Response 2.6: We have now added new figures (Fig. R2.3) to the supplementary information (Supplementary Fig. 8), showing the rescaled Figure 3d-f. The intensity ratio of the SHG in co- and cross-polarization configurations is roughly 1:12, corresponding to a helicity imperfection of approx. 8.3 % arising from the instrument response. This imperfection could come from both the excitation and the detection pathway.

Figure R2.3 | Polarization-resolved SHG and UPL spectra from monolayer WSe_2 at gate voltages of 2 V (a), 0 V (b), and -2 V (c). These spectra are identical to Fig. 3d-f but shown on a different scale.

We have also followed the referee's suggestion and added a color plot of the degree of helicity of the neutral and charged HX emission from MoSe_2 to the supplementary information (new Supplementary Fig. 8), also shown below in Fig. R2.4.

Figure R2.4 | Helicity of the UPL of the high-lying trions from monolayer MoSe_2 as a function of the gate voltage.

7. During the data analysis of fig4, the existence of the dip and the dip position of the SHG signal is critical to understand the quantum-interference feature in the resonant SHG of the coherently driven system. While in the fig 4c, the 1.7eV laser is very close to the exciton resonance frequency. In the color plot of fig 4c, will UPL and SHG contribute together to the spectra? If UPL also contributes to the spectra, the data analysis of the SHG will be problematic and the existence of the dip and the dip position will be not convincing.

Response 2.6: These are indeed fair concerns. The A-exciton is at 1.725 eV (see Fig. 1d) and shifts to higher energies at higher carrier densities. The UPL intensity is more than 10 times weaker than the SHG under excitation by an 80-fs pulsed laser. We have added a gray line in Fig. 4d (Fig. R2.5 below) to indicate the UPL of the *s*-like HX around an energy of 3.325 eV, matching with the result from CW excitation shown in Fig. 2e. The intensity of the UPL (gray line) is magnified tenfold and shifted on the y-axis for clarity. Therefore, due to the difference in spectral position and the intensity being more than an order of magnitude lower, we can safely rule out contributions from the UPL to the dips in the SHG spectrum.

Figure R2.5 | SHG spectra at gate voltages of 2.4 V, 0 V, and -2.4 V. Black arrows mark the SHG dips due to excitonic quantum interference. The gray line shows a part of the same spectrum at 0 V, multiplied by 10 and shifted along the y-axis, revealing the UPL of the s-like HX under femtosecond pulsed excitation.

Reviewers' Comments:

Reviewer #1:

Remarks to the Author:

Following the advices from the referees, the authors have performed considerable improvements in the new version of the manuscript. They have addressed mostly my concern/questions. The manuscript is ready for publication now, I think.

Reviewer #2:

Remarks to the Author:

In this revision, the authors satisfactorily addressed all my technical remarks and concerns as well as the theoretical interpretations. The added discussion and data have improved the overall quality of the manuscript. Though the theory part of this work is approximation to support the experiment, the experiment result is solid, and the experiment evidence is strong. In view of comments from the other referee and corresponding authors' rebuttal, I would recommend this paper to be published on Nature Communications. However, I suggest the author to either improve the gate voltage resolution or remove the Supplementary Fig 8d, as the resolution of the current figure is too low.

RESPONSE TO REVIEWER COMMENTS

Reviewer #1 (Remarks to the Author):

Following the advices from the referees, the authors have performed considerable improvements in the new version of the manuscript. They have addressed mostly my concern/questions. The manuscript is ready for publication now, I think.

Response: We thank the referee for providing valuable advice that helped us to improve the manuscript and for supporting our publication.

Reviewer #2 (Remarks to the Author):

In this revision, the authors satisfactorily addressed all my technical remarks and concerns as well as the theoretical interpretations. The added discussion and data have improved the overall quality of the manuscript. Though the theory part of this work is approximation to support the experiment, the experiment result is solid, and the experiment evidence is strong. In view of comments from the other referee and corresponding authors' rebuttal, I would recommend this paper to be published on Nature Communications. However, I suggest the author to either improve the gate voltage resolution or remove the Supplementary Fig 8d, as the resolution of the current figure is too low.

Response: We thank the reviewer for the constructive remarks and for recommending our publication. On the advice of the reviewer, we have removed supplementary Fig. 8d.